# The Genus *Psenulus* Kohl, 1897 (Hymenoptera: Crabronidae) in China, with Two New Species and Two New Country Records [note 1]

**DOI:** 10.3390/insects16040432

**Published:** 2025-04-19

**Authors:** Huifen Jiang, Qiang Li, Li Ma

**Affiliations:** Department of Entomology, College of Plant Protection, Yunnan Agricultural University, Kunming 650201, China; jianghf0538@163.com

**Keywords:** Pemphredoninae, *Psenulus*, China, new species, key

## Abstract

*Psenulus* Kohl, 1897 is the largest genus in Pemphredoninae and has a cosmopolitan distribution. In this study, we first review the genus *Psenulus* from China and discover two new species and two new records from China. We comprehensively analyse and document intraspecific morphological variations and give identification keys to females and males of *Psenulus* species from China.

## 1. Introduction

The genus *Psenulus* Kohl, 1897 belongs to the tribe Psenini, subfamily Pemphredoninae (Hymenoptera: Crabronidae). It was established by Kohl in 1897 [1], with the type species *Mimesa fuscipennis* Dahlbom [=*Psen fuscipennis* Dahlbom, 1843] which was designated by Ashmead in 1899 [2]. *Psenulus* is the largest genus in Pemphredoninae with 173 species and 50 subspecies, represented in all the World Zoogeographic Regions. Of these, 93 species and 39 subspecies occur in the Oriental region, 32 species and 3 subspecies in the Ethiopian region, 24 species and 3 subspecies in the Palearctic region, 12 species in the Neotropical region, 3 species and 3 subspecies in the Australasian region, 3 species and 1 subspecies in the Nearctic region, 3 species in both the Palearctic and Oriental regions, 2 species and 1 subspecies in the Oriental and Australasian regions, and 1 species across the Nearctic–Palearctic–Oriental–Neotropical regions [3]. To date, 16 species and 5 subspecies of *Psenulus* have been recorded from China [3,4,5,6,7].

*Psenulus* consists mostly of small- to medium-sized predatory wasps, with a body length of 4–12 mm. They capture and paralyze insects, such as Aphididae, Cicadellidae, Psyllidae, and Delphacidae, both nymphs and adults [3,4,8].

In *psenulus*, adult females use silk-like materials to line the nest walls and make partitions between brood cells [9,10,11]. Melo examined the structures associated with silk production in 33 species of *Psenulus* and recognized that the source of the silk secreted from glands was associated with gastral sterna V and VI, where spinnerets formed by rows of hollow setae were located [8,11]. Serrao and Oliveira described the structure of silk glands in *Psenulus* using light and scanning electron microscopy [12]. Their findings confirmed that the silk glands in *Psenulus* evolved independently [11,12].

In the present work, we provide the first systematic review of the genus *Psenulus* in China, as no comprehensive taxonomic study has been conducted on the Chinese species of *Psenulus*. This taxonomic study of *Psenulus* led to the discovery of two new species and two new records from China. The new species are described and the new records are given supplementary descriptions. We also discuss the intraspecific morphological variations in the species examined. Additionally, keys to females and males of *Psenulus* species from China are provided.

## 2. Materials and Methods

The specimens examined were obtained from CAU, NWAFU, YNAU, ZAFU, and ZJU, and they are all deposited in the Insect Collection of Yunnan Agricultural University, Kunming, P. R. China (YNAU).

CAU: China Agricultural University, Beijing, China.

NWAFU: Northwest Agriculture and Forestry University, Yangling, Shanxi, China.

YNAU: Yunnan Agricultural University, Kunming, Yunnan, China.

ZAFU: Zhejiang Agriculture and Forestry University, Hangzhou, Zhejiang, China.

ZJU: Zhejiang University, Hangzhou, Zhejiang, China.

Adult external morphology was studied under the Olympus stereomicroscope (SZ Series) (Olympus Corporation, Tokyo, Japan) with an ocular micrometer. High-quality photos were taken with Keyence VHX-5000 (Keyence Corporation, Osaka, Japan). All images were post-processed using Adobe Photoshop 2020.

Morphological terminology follows that of Bohart and Menke 1976 [4], and van Lith 1962 [13]. Abbreviations in the text are as follows: “n” refers to the number of samples; HL = head length in frontal view; HW = head width in frontal view; PL = petiole length in dorsal view; POD = postocellar distance; OOD = ocellocular distance.

## 3. Results

### 3.1. Systematics

*Psenulus* Kohl, 1897

*Psenulus* Kohl, 1897: 293 [1]. Type species: *Mimesa fuscipennis* Dahlbom [=*Psen fuscipennis* Dahlbom, 1843], designated by Ashmead, 1899: 224 [2].

*Neofoxia* Viereck, 1901: 338 [14]. Type species: *Psen atratus* of Panzer, 1806 [=*Trypoxylon atratum* Fabricius, 1804 = *Sphex pallipes* Panzer, 1798], by original designation.

*Stenomellinus* W. Schulz, 1911: 142 [15]. Type species: *Psen dilectus* de Saussure, 1892, by monotypy.

*Eopsenulus* Gussakovskij, 1934: 84 [16]. Type species: *Psenulus iwatai* Gussakovskij, 1934, by original designation.

Diagnosis. Antennal sockets placed well above frontoclypeal suture; midtibia with single spur; forewing with three submarginal cells; hindwing media diverging after vein cu-a; frontal carina elevated between antennal sockets, dorsal surface of projecting part sharp or broadened, usually meeting a transverse carina below sockets; anterior oblique suture usually foveolate; petiole in dorsal view generally smooth and shiny [4].

Distribution. Cosmopolitan [3].

Key to the species of the genus *Psenulus* Kohl from China:

Female

Female unknown for *P. hoozanius* van Lith, 1972; *P. ornatus ornatus* (Ritsema), 1876; *P. ornatus kankauensis* Strand, 1916; *P. quadridentatus formosanus* Tsuneki, 1966.
1. Interantennal carina prominent, dorsally sharp……………………………………………2- Interantennal carina prominent, dorsally broadened and concave………………………...82. Thorax black…………………………*P. ephippius* Taylor, Barthélémy, Chi, and Guénard- Thorax black with yellow marks or almost entirely yellow…………………………………33. Thorax almost entirely yellow…………*P. pallens* Taylor, Barthélémy, Chi, and Guénard- Thorax black with yellow marks……………………………………………………………....44. Petiole black; gaster black, segments II with reddish-brown marks to various degrees; transverse carina below antennae absent…………………*P. carinifrons rohweri* van Lith- Petiole yellow to red at base, dark-brown distally; gaster yellow to red; transverse carina below antennae distinct or absent……………………………………………………………55. Transverse carina below antennae absent; pygidial plate elongate-triangular with sharp lateral carinae…………………………………………………………*P. continentis* van Lith- Transverse carina below antennae conspicuously long; pygidial plate short and narrow with distinct or indistinct lateral carinae…………………………………………………….66. Free margin of clypeus with two blunt triangular teeth medially, emargination between teeth blunt triangle; lateral mark of mesoscutum along tegula short and narrowly band-shaped, not reaching anterior margin of mesoscutum…*P. maculatus maculatus* van Lith- Free margin of clypeus slightly or conspicuously prominent and truncate medially; lateral mark of mesoscutum along tegula reaching anterior margin of mesoscutum…77. Free margin of clypeus with rectangular prominence medially, deeply emarginate on either side of prominence; pygidial plate with distinct lateral carinae, parallel on apical half; median marks of mesoscutum band-shaped, as long as mesoscutum, ending in one large, transverse mark along mesoscutum posterior margin..*P. xanthonotus* van Lith- Free margin of clypeus with slightly truncate prominence medially; pygidial plate with indistinct lateral carinae; median marks of mesoscutum long and narrowly stripe-shaped, reaching anterior margin of mesoscutum and not extending to posterior margin………………………………………...……………*P. ornatus pempuchiensis* Tsuneki8. Gaster yellow to red………………………………………………………………………….9- Gaster black or mostly black…………………………………………………………………149. Free margin of clypeus quadridentate medially………………………………………….10- Free margin of clypeus bidentate or truncate medially……………………………………1110. Free margin of clypeus quadridentate medially, small, of same size and at same horizontal level; pygidial plate narrowly grooved, lateral carinae almost parallel………………………………………………...……………...*P. dentideus* Ma and Li- Free margin of clypeus bidentate medially, deeply emarginate on either side of teeth and with small tooth on outer side of emargination; pygidial plate elongate-triangular with conspicuous lateral carinae………………………………………*P. quadridentalus* van Lith11. Body length < 9 mm, medium-sized; propodeal enclosure incomplete or short; dorsal surface of petiole without longitudinal carina medially…………………………………12- Body length > 10 mm, large-sized; propodeal enclosure complete and long; dorsal surface of petiole with longitudinal carina medially………………………………………………1312. Petiole black; transverse carina below antennae absent; propodeal enclosure narrowly triangular, with median longitudinal carina; base of hind tibia with longitudinal groove……………………………………*P. gibbus* Taylor, Barthélémy, Chi, and Guénard- Petiole reddish-yellow; transverse carina below antennae conspicuously long, arched in frontal view; propodeal enclosure short, triangular, without median longitudinal carina; hind tibia without longitudinal groove…………………………*P. parvidentatus* van Lith13. Petiole dark reddish-brown, almost black; transverse carina below antennae absent; propodeal enclosure well-defined posteriorly…………………*P. clypeoconvexus* sp. nov.- Petiole reddish-yellow; transverse carina below antennae conspicuously long, almost reaching outer side of antennal sclerites; propodeal enclosure ill-defined posteriorly……………………………………………………………*P. carinitibialis* sp. nov.14. Transverse carina below antennae absent; pygidial plate absent…*P. suifuensis* van Lith- Transverse carina below antennae conspicuously long; pygidial plate with distinct or indistinct lateral carinae……………………………………………………………………1515. Petiole long; propodeal enclosure ill-defined posteriorly; propodeum behind enclosure smooth and shiny…………………….………………………………………………………16- Petiole short; propodeal enclosure well-defined or indistinctly defined posteriorly; propodeum behind enclosure rugose………………………………………………………1716. Gaster black with some segments red; transverse carina below antennae conspicuously long, almost reaching outer side of antennal sclerites; dorsal surface of petiole with broad longitudinal excavation, lateral surface somewhat carinated…*P. bicinctus* Turner- Gaster black, lateral surface somewhat brown; transverse carina below antennae short, reversed V-shaped; dorsal surface of petiole with deep longitudinal groove, lateral surface with sharp carina at base…………………………………………*P. orinus* van Lith17. Dorsal surface of petiole with broad longitudinal depression, ventral surface rounded; pygidial plate broadly triangular………………………………………………….……….18- Dorsal surface of petiole with longitudinal groove medially, ventral surface carinate; pygidial plate long and narrowly groove-shaped………………………………………1918. Free margin of clypeus with deep emargination medially, subtriangular; pygidial plate with indistinct lateral carinae; propodeum behind enclosure with fine, oblique longitudinal striation………………………………………………………*P. pallipes* (Panzer)- Free margin of clypeus with relatively shallow, arcuate emargination medially; pygidial plate with conspicuously sharp lateral carinae; propodeum behind enclosure coarsely carinate and rugose……………………………………………………...*P. formosicola* Strand19. Mandible tridentate, middle tooth broad and blunt; central part of clypeal margin with shallow depression; pygidial plate narrowly grooved…………………*P. lubricus* (Pérez)- Mandible tridentate, middle tooth narrow and sharp; central part of clypeal margin deeply impressed; pygidial plate elongated and linear………………*P. yingfeng* Tsuneki

Male

Male unknown for *P. parvidentatus* van Lith, 1972; *P. taihorinis* Strand, 1916; *P. xanthonotus* van Lith, 1969; *P. yingfeng* Tsuneki, 1982
1. Antennae without tyloids……………………………………………………………………2- Antennae with tyloids………………………………………………………………………...112. Thorax black…………………………………………………………………………………...3- Thorax yellow or thorax black with yellow marks…………………………………………63. Gaster black; petiole short, subquadrate, dorsal surface of petiole with longitudinal groove medially…………………………………………………………*P. formosicola* Strand- Gaster yellow to red; petiole short, cylindrical, dorsal surface smoooth and shiny…….44. Dorsal surface of propodeum smooth and shiny, rest of propodeum densely and coarsely reticulate; interantennal carina sharply elevate……………………………………*P. ephippius*
Taylor, Barthélémy, Chi, and Guénard- Apex of posterior surface and lateral surface of propodeum coarsely reticulate; interantennal carina slightly broadened but not excavate…………………………………55. Propodeal enclosure well-defined posteriorly, propodeum behind enclosure shiny and rugose; free margin of clypeus with semicircular emargination medially…………………………………………………………*P. clypeoconvexus* sp. nov.- Propodeal enclosure ill-defined posteriorly, propodeum behind enclosure smooth and shiny; free margin of clypeus with triangular emargination medially……………………………………………………………*P. carinitibialis* sp. nov.6. Mesothorax almost entirely yellow; free margin of clypeus with two sharp triangular teeth medially, emargination between teeth sharply triangular………………………………*P. pallens* Taylor, Barthélémy, Chi, and Guénard- Mesothorax black with yellow marks; free margin of clypeus slightly convex and with two inconspicuous teeth medially or distinctly prominent and truncate medially……77. Petiole black; gaster black; free margin of clypeus with distinctly truncate prominence medially………………………………………………………*P. carinifrons rohweri* van Lith- Petiole yellow to red, except black at apex; gaster yellow to red; free margin of clypeus slightly convex, with two faint teeth medially……………………………………………...88. Propodeum yellow except enclosure and median longitudinal groove black; mesoscutum with large yellow marks, broadly band-shaped……*P. continentis* van Lith- Propodeum black with yellow marks on posterior surface; mesoscutum with small yellow marks, band-shaped or stripe-shaped or patchy…………………………………99. Lateral mark of mesoscutum along tegula short and narrowly band-shaped, median marks of mesoscutum at apex triangularly patchy…………………*P. hoozanius* van Lith- Mesoscutum along tegula with or without mark, median marks long and narrowly stripe-shaped, broadened at apex. …………………………………………………………1010. Mesoscutum along tegula without mark, median marks beginning at prescutal sutures and not reaching anterior margin of mesoscutum; free margin of clypeus slightly convex, almost truncate…………………………………………*P. ornatus ornatus* (Ritsema)- Mesoscutum along tegula with yellow mark, median marks reaching anterior margin of mesoscutum; free margin of clypeus distinctly prominent……………………………………………………*P. ornatus pempuchiensis* Tsuneki11. Gaster yellow to red………………………………………………………………………...12- Gaster black or mostly black…………………………………………………………………1512. Thorax black with yellow marks; interantennal carina sharp…………………………………………………………*P. maculatus maculatus* van Lith- Thorax black; interantennal carina broadened dorsally, concave or not…………………1313. Petiole black; interantennal carina broadened dorsally but not concave; mesopleural suture smooth, not foveolate……………*P. gibbus* Taylor, Barthélémy, Chi, and Guénard- Petiole yellow to red; interantennal carina broadened dorsally and concave; mesopleural suture foveolate………………………………………………………………………………1414. Gaster yellow; flagellomeres narrowly rectangular, with short carina-like tyloids; first recurrent vein received by first submarginal cell………………*P. quadridentalus* van Lith- Gaster red; flagellomeres stubby and rounded, with elongated oval tyloids; first recurrent vein received by first submarginal cell, second submarginal cell, or interstice between first and second submarginal cells………………………*P. dentideus* Ma and Li15. Transverse carina below antennae absent or inconspicuous………*P. suifuensis* van Lith- Transverse carina below antennae conspicuously long……………………………………1616. Flagellomeres IV-VIII with linear tyloids; propodeum behind enclosure carinate and rugose………………………………………………………………………*P. pallipes* (Panzer)- Flagellomeres I-X or I-XI with carina-like tyloids; propodeum behind enclosure smooth and shiny……………………………………………………………………………………1717. Flagellomeres with short carina-like tyloids, first and last with tuberculate tyloids; free margin of clypeus with shallow emargination medially………………...*P. lubricus* (Pérez)- Flagellomeres with long, oblique, narrow carina-like tyloids, last one reduced to very short carina or point; free margin of clypeus with relatively deep emargination medally………………………………………………………………………………………1818. Gaster black with red segments; transverse carina below antennae conspicuously long but not reaching outer side of antennal sclerites………………………*P. bicinctus* Turner- Gaster black; transverse carina below antennae conspicuously long, reaching outer side of antennal sclerites………………………………………………………*P. orinus* van Lith

### 3.2. Description of New Species

#### 3.2.1. *Psenulus carinitibialis* Jiang & Ma, sp. nov. (Figure 1 and Figure 2)

Material examined. Holotype, ♀, China: Yunnan, Xishuangbanna, Menghai County, Bulang Mountain, 21°44′ N, 100°26′ E, alt. 1659 m, 21.VI-20.VII.2018, Malaise trap, coll. Huifen Jiang (YNAU). Paratypes. 62♀♀13♂♂, same location as holotype, 17.V-21.VI.2018 (9♀♀), 21.VI-20.VII.2018 (4♀♀), 20.VII-15.VIII.2018 (1♀), 19.IV-28.V.2019 (4♀♀5♂♂), 28.V-28.VI.2019 (20♀♀), 28.VI-19.VII.2019 (9♀♀), 19.VII-21.VIII.2019 (3♀♀), 11.VII-13.VIII.2020 (1♀), 15.VIII-12.IX.2020 (1♀), 14.III-15.IV.2021 (1♂), 15.IV-27.V.2021 (5♂♂), 27.V-15.VI.2021 (9♀♀1♂), 16.VII-16.VIII.2021 (1♀), 16.VIII-15.IX.2021 (1♂), Malaise trap, coll. Huifen Jiang (YNAU); 3♀♀, China: Yunnan, Xishuangbanna, Menghai County, Guanggang Village, 21°49′ N, 100°29′ E, alt. 1559 m, 20.III-22.IV.2019 (2♀♀), 2-22.VII.2019 (1♀), Malaise trap, coll. Huifen Jiang (YNAU); 1♀, China: Henan, Jigong Mountain, alt. 700 m, 11.VII.1997, Chikun Yang (CAU).

Diagnosis. This species shares the following with *P. pendleburyi* van Lith, 1962 [13], of which the male is unknown: body size larger than average in *Psenulus*; interantennal carina markedly prominent, dorsally broadened, and deeply concave; outer surface of mid tibia with longitudinal carina medially and posteriorly, margined behind by row of stout spines; pygidial plate elongate-triangular, lateral carinae parallel apically; prominent margin of clypeus with two teeth; frons below transverse carina with shallow circular depression; same colour pattern. Markedly differs by the characters given in the Table 1.

This species is unique in having the following character combination, which distinguishes it from all congeners: complete and sharp transverse carina below antennae, outer surface of mid tibia with sharp longitudinal carina medially and posteriorly and without depression at apex, hind tibia near base with longitudinal groove and longitudinal lamina carina along the groove.

Description. Female (*n* = 67). Body length 10.1–12.2 mm. Holotype:

Colour. Head and thorax black (Figure 1A); mandible except dark tip, palpi and tegula yellow (Figure 1A,B,D); labrum rufous; scape beneath yellow, fulvous dorsally, rest of antennae reddish-brown ventrally, dark-brown dorsally (Figure 1A–C); pronotal lobe pale yellow to yellow (Figure 1F); veins of wings brown (Figure 1A); coxae dark brown basally, rest of legs pale yellow to reddish-yellow (Figure 1A); spines on mid and hind tibia rufous; petiole and gaster reddish-yellow (Figure 1A).

**Figure 1 insects-16-00432-f001:**
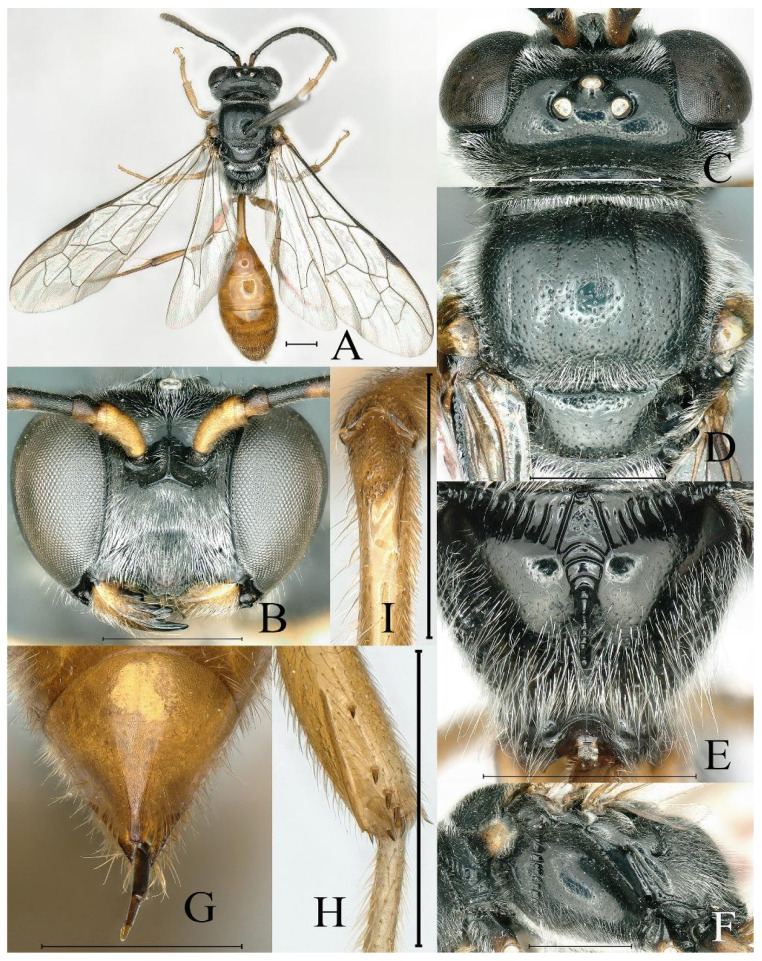
*Psenulus carinitibialis* Jiang & Ma, sp. nov., female, holotype: (**A**,**C**,**G**); paratype: (**B**,**D**–**F**,**H**,**I**). (**A**) Habitus, dorsal view; (**B**) head, frontal view; (**C**) head, dorsal view; (**D**) throax, dorsal view; (**E**) propodeum, dorsal view; (**F**) thorax, lateral view; (**G**) pygidial plate, dorsal view; (**H**) mid tibia medially and posteriorly, lateral view; (**I**) base of hind tibia, dorsal view. Scale bar = 1 mm.

Pubescence. Mandibles, prominent margin of clypeus, petiole and gaster with golden pubescence (Figure 1A,B,G); lower part of epicnemial area with round patch of appressed pubescence; gastral sterna V and VI with apical fringe of short silvery hairs; remaining of body with silvery pubescence.

Head. Mandible bidentate at apex and with one distinct inner tooth in middle (Figure 1B); labrum nearly rectangular, anterior margin with two triangular teeth; prominent margin of clypeus with two blunt triangular teeth, emargination between teeth broad and shallow (Figure 1B); interantennal carina conspicuously prominent, dorsally broadened and deeply concave (Figure 1B,C); transverse carina below antennae elevated sharply, its ends bent upwards to outer side of antennal sclerites (Figure 1B); frons below interantennal carina with shallow circular, ill-defined depression (Figure 1B); upper frons densely, finely punctate; ocellar triangle slightly raised, sparsely and finely punctate, outer side of posterior ocelli slightly depressed (Figure 1C); vertex behind ocelli finely, sparsely punctate, broadly and deeply concave posteriorly (Figure 1A,C); gena sparsely, minutely punctate; occipital carina not reaching hypostomal carina. HL = 1.9 mm; HW = 2.2 mm; POD: OOD = 14: 22.

Thorax. Pronotal collar with sharp marginal carina and inconspicuous lateral anterior angles (Figure 1A,D); mesoscutum shiny, anteriorly and laterally with mid-sized and dense punctures, posteriorly very densely punctate, rest with fine and coarse punctures intermixed (Figure 1A,D); prescutal sutures distinctly grooved, extending to 2/3 of scutum length (Figure 1A,D); admedian line and parapsidal line slightly elevated as smooth carinae (Figure 1A,D); scutellum shiny, with mid-sized and sparse punctures (Figure 1A,D); metanotum with fine and dense punctures (Figure 1A,D); propodeal enclosure triangular, not defined posteriorly, and with distinct oblique longitudinal carinae (Figure 1A,E); median longitudinal groove narrow, deep, almost reaching apex of propodeum (Figure 1E); dorsal surface and most posterior surface of propodeum smooth and shiny, lateral area and apex of posterior surface and lateral surface of propodeum with irregular carinations and rugae (Figure 1E,F); episternal sulcus conspicuously foveolate (Figure 1F); mesopleural suture with narrow fovea on basal half (Figure 1F); mesopleuron and metapleuron smooth, shiny, mesopleuron finely, sparsely punctate, metapleuron almost impunctate (Figure 1F).

Wings. First recurrent vein received by first submarginal cell, second recurrent vein received by third submarginal cell (Figure 1A).

Legs. Ventral surface of fore trochanter truncate, with distinct, sharp lateral carina; outer surface of mid tibia with sharp longitudinal carinae medially and posteriorly, margined behind by row of five to seven rufous stout spines and two spines at apex (Figure 1H); hind tibiae near base with longitudinal groove and longitudinal lamina carina along groove, on top of groove with many rufous tuberculate spines (Figure 1I); inner spur of hind tibia shorter than basitarsus (Figure 1A). 

Gaster. Petiole cylindrical, smooth and shiny, dorsal surface of petiole with inconspicuous longitudinal carina medially, ending in triangular concavity (Figure 1A); PL = 2.0 mm; base of sternum II with ill-defined, U-shaped depression; tergum VI inconspicuously coriaceous (Figure 1G); pygidial plate elongate-triangular, distinctly coriaceous, and with sharp lateral carinae, carinae almost parallel apically, extending for approximately 1/3 of pygidial plate length (Figure 1G).

Male (*n* = 13). Body length 9.7–10.6 mm. Mandible except dark tip, palpi, scape, ventral surface of pedicel, tegula, legs except coxae dark brown basally, petiole and gaster yellow to reddish-yellow (Figure 2A); dorsal surface of scape and pedicel yellowish-brown (Figure 2A,C); flagellum beneath reddish-yellow to fulvous and brown towards apex, dorsally black; lower part of epicnemial areas without circular patch of appressed hairs; mandible without inner tooth (Figure 2B); prominent margin of clypeus with two distinct triangular teeth, emargination between teeth triangular (Figure 2B); antenna moniliform, without tyloids (Figure 2A); interantennal carina sharp, slightly or indistinctly broadened and not excavate dorsally (Figure 2B,C); HL = 1.4–1.7 mm, HW = 1.7–2.2 mm, POD: OOD = 13–14: 17–20; lateral area and apex of posterior surface and lateral surface of propodeum coarsely reticulate (Figure 2E,F); legs usual; petiole usual, PL = 1.4–1.7 mm. Other characters as in female.

Distribution. China (Henan, Yunnan).

**Figure 2 insects-16-00432-f002:**
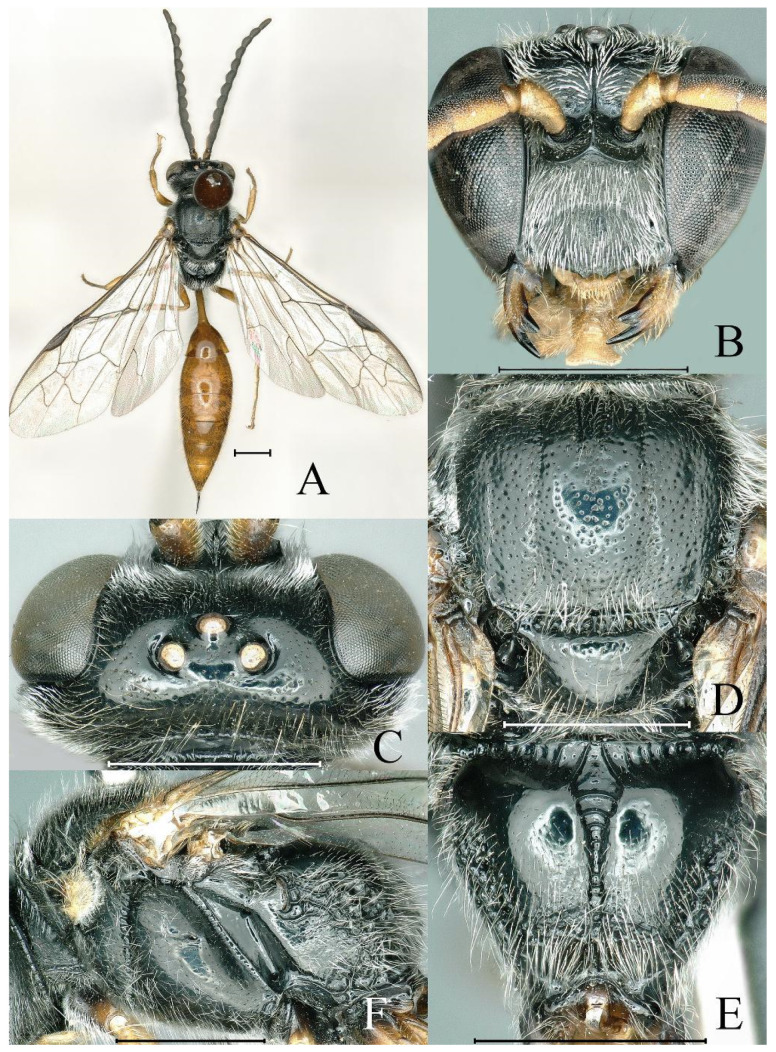
*Psenulus carinitibialis* Jiang & Ma, sp. nov., male, paratype. (**A**) Habitus, dorsal view; (**B**) head, frontal view; (**C**) head, dorsal view; (**D**) throax, dorsal view; (**E**) propodeum, dorsal view; (**F**) thorax, lateral view. Scale bar = 1 mm.

Etymology. The name *carinitibialis*, derived from the Latin words *carin* (=carina) and *tibialis* (=tibial), referring the mid and hind tibiae with longitudinal carina, one of the main recognition characters of this species.

Notes. This species belongs to the *Psenulus quadridentatus* group.

The paratypes vary as follows:

Variation in colouration. Female: labrum fulvous in some specimens; fore and mid legs with ivory area to various degrees, hind leg ivory to yellow; petiole ranging from transparent pale yellow to reddish-brown; base half of gastral sternum II with triangular ivory patch, upper part of gastral sterna V and VI with ivory patches to various degrees; male: fore and mid legs laterally with dark-brown patches; gaster dull in colour, ranging from yellow to dark brown. Overall, the leg and gaster colouration of paratypes varies from pale yellow to yellow to reddish-yellow. In addition, one female from Henan Province is relatively dark, particularly with deep reddish-brown gaster. Apart from this, its characteristics agree with the holotype, so we identify it as *P. carinitibialis* sp. nov.

Variation in forewing venation. First recurrent vein received by first submarginal cell in most specimens examined, in some specimens received by interstice between first and second submarginal cells, and in some specimens received differently in the left wing and the right wing. As the forewing venation is unstable, it cannot be a recognition character of this species. 

#### 3.2.2. *Psenulus clypeoconvexus* Jiang & Ma, sp. nov. (Figure 3 and Figure 4)

Material examined. Holotype, ♀, China: Zhejiang, Hangzhou, Xixi National Wetland Park, VI–VII.2014, Gang Yao, Malaise trap (YNAU). Paratypes. 2♀♀, same to holotype; 1♀, China: Zhejiang, Hangzhou, Shunxiwu, Qingliang Peak, 24–25.VI.2011 (ZAFU); 1♂, China: Zhejiang, Linan, West Tianmu Mountain, 1.VII.2000, Xuexin Chen (YNAU); 1♂, China: Zhejiang, Linan, West Tianmu Mountain, Xianren Peak, alt. 1520 m, 1.VII.2000, Meihua Piao (YNAU); 1♀, China: Guangxi, Guilin, Yanshan Botanical Garden, alt. 154 m, 30.VII–14.VIII.2020, Jingxia Gao (YNAU).

Diagnosis. This species resembles *P. carinitibialis* sp. nov. in the following characters: body size larger than average in *Psenulus*; same colour-pattern except petiole; prominent margin of clypeus with two teeth; female: interantennal carina markedly prominent, dorsally broadened and concave; hind tibia near base with longitudinal groove; male: antennae without tyloids; lateral area and apex of posterior surface and lateral surface of propodeum coarsely reticulate. Markedly differs by the characters given in the Table 2.

Female of this species has no transverse carina below antennae, mid tibiae at apex with flatten longitudinal depression, and hind tibia basally with longitudinal shallow groove, which distinguishes this species from all congeners.

Description. Female (*n* = 5). Body length 11.6–12.5 mm. Holotype: 

Colour. Head, thorax and coxae black (Figure 3A); mandibles except dark tips, palpi and tegula yellow (Figure 3A,B); labrum rufous; scape beneath pale yellow, fulvous dorsally, rest of antennae reddish-yellow ventrally, blackish-brown dorsally (Figure 3A–C); pronotal lobe ivory to pale yellow (Figure 3F); wing veins brown (Figure 3A); fore and mid legs except coxae pale yellow to yellow, ivory to pale yellow ventrally, outer lower edge of mid femur with brown patches (Figure 3A); hind leg except coxa reddish-yellow, outer margin of knees brown (Figure 3A); petiole dark reddish-brown, almost black (Figure 3A); gaster reddish-yellow (Figure 3A,G).

**Figure 3 insects-16-00432-f003:**
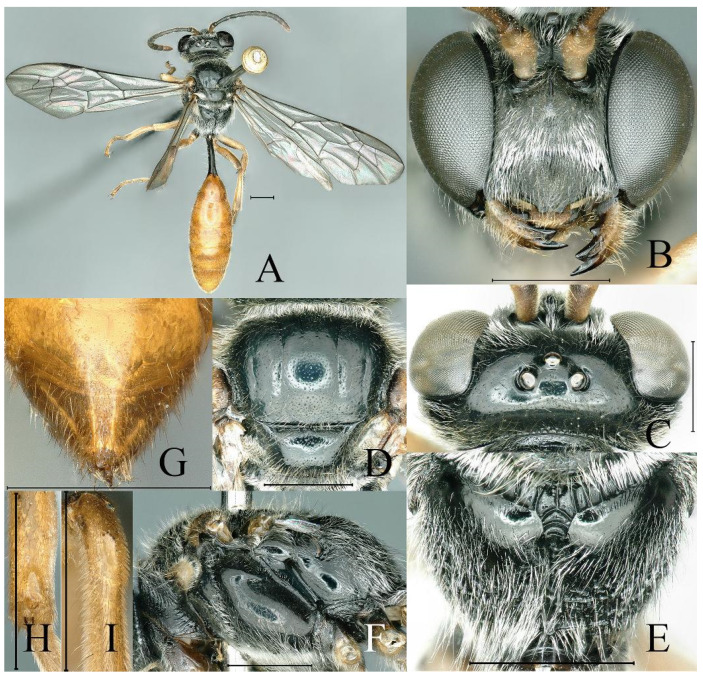
*Psenulus clypeoconvexus* Jiang & Ma, sp. nov., female, holotype: (**A**,**B**,**C**,**E**,**G**–**I**); paratype: (**D**,**F**). (**A**) Habitus, dorsal view; (**B**) head, frontal view; (**C**) head, dorsal view; (**D**) throax, dorsal view; (**E**) propodeum, dorsal view; (**F**) thorax, lateral view; (**G**) pygidial plate, dorsal view; (**H**) mid tibia posteriorly, lateral view; (**I**) base of hind tibia, dorsal view. Scale bar = 1 mm.

Pubescence. Mandibles and gaster with golden pubescence (Figure 3A,B,G); lower part of epicnemial area with round patch of dense appressed pubescence; gastral sterna V and VI with apical fringe of silvery short hairs; rest of body with silvery pubescence.

Head. Mandible bidentate at apex and with distinct inner tooth in middle (Figure 3B); labrum nearly rectangular, anterior margin truncate; prominent margin of clypeus with two blunt triangular teeth, emargination between teeth nearly truncate (Figure 3B); interantennal carina markedly prominent, in dorsal view markedly broadened and with oval, shallow depression (Figure 3B,C); frons below antennae conspicuously raised and without distinct transverse carina (Figure 3B); frons below interantennal carina with inconspicuous, shallow circular depression (Figure 3B); upper frons densely, finely punctate; ocellar triangle slightly raised, sparsely, finely punctate, outer side of posterior ocelli slightly depressed (Figure 3C); vertex behind ocelli finely, sparsely punctate, broadly and deeply concave posteriorly (Figure 3C); gena sparsely, minutely punctate; occipital carina reaching hypostomal carina. HL = 2.0 mm; HW = 2.6 mm; POD:OOD = 17:25.

Thorax. Pronotal collar with sharp marginal carina and distinct lateral anterior angle; mesoscutum shiny, fine to mid-sized punctures intermixed, distributed relatively evenly (Figure 3D); prescutal sutures distinctly grooved, extending to 3/4 of scutum length (Figure 3A,D); admedian line slightly elevated as smooth carina (Figure 3D); parapsidal line concave (Figure 3D); scutellum shiny, with sparse mid-sized punctures (Figure 3D); metanotum with fine and dense punctures (Figure 3D,E); propodeal enclosure crescent-shaped, defined posteriorly and with distinct oblique longitudinal carinae (Figure 3A,E); behind enclosure shiny and rugose (Figure 3E); median longitudinal groove narrow and deep, almost reaching apex of propodeum (Figure 3E); dorsal surface and upper part of posterior surface of propodeum smooth and shiny, most posterior surface and lateral surface of propodeum with relatively coarse and irregular reticulation (Figure 3E,F); episternal sulcus conspicuously foveolate (Figure 3F); mesopleural simple, without fovea (Figure 3F); mesopleuron and metapleuron smooth and shiny, mesopleuron finely, sparsely punctate, metapleuron almost impunctate (Figure 3F).

Wings. First recurrent vein received by second submarginal cell, second recurrent vein received by third submarginal cell (Figure 3A).

Legs. Ventral surface of fore trochanter and fore femur truncate, with blunt lateral carinae; mid tibia with flatten longitudinal depression apically, and two rufous spines at apex (Figure 3H); hind tibia near base with longitudinal shallow depression and longitudinal carina along depression, on top of depression with few rufous tuberculate spines (Figure 3I); inner spur of hind tibia shorter than basitarsus.

Gaster. Petiole cylindrical, smooth, shiny, dorsal surface of petiole with inconspicuous longitudinal carina medially, ending in triangular concavity (Figure 3A); PL = 2.0 mm; base of gastral sternum II with ill-defined, U-shaped depression; pygidial plate triangularly elongate, finely coriaceous, lateral carinae blunt and almost parallel (Figure 3G).

Male (*n* = 2). Body length 10.0–10.3 mm. Wing veins light brown (Figure 4A); legs except coxae reddish-yellow, femora with brown patches to various degrees; upper half of gastral tergum IV, upper part of gastral sterna III to V with brown patches (Figure 4A); lower part of epicnemial area without circular patch of appressed hairs; mandible without inner tooth (Figure 4B); prominent margin of clypeus with two distinct triangular teeth, emargination between teeth subsemicircular (Figure 4B); antenna moniliform, without tyloids (Figure 4A); interantennal carina sharp, slightly or indistinctly broadened and not excavate dorsally (Figure 4B,C); HL = 1.5–1.6 mm, HW = 1.9–2.1 mm, POD: OOD = 13–14: 18–19; mesoscutum surface relatively coarse, with dense mid-sized to large punctures (Figure 4A); mesopleural suture with indistinct small fovea (Figure 4E); first recurrent vein received by first submarginal cell (Figure 4A); legs usual; base of petiole with flatten longitudinal depression (Figure 4A); PL = 1.2–1.4 mm. Other characters as in female.

**Figure 4 insects-16-00432-f004:**
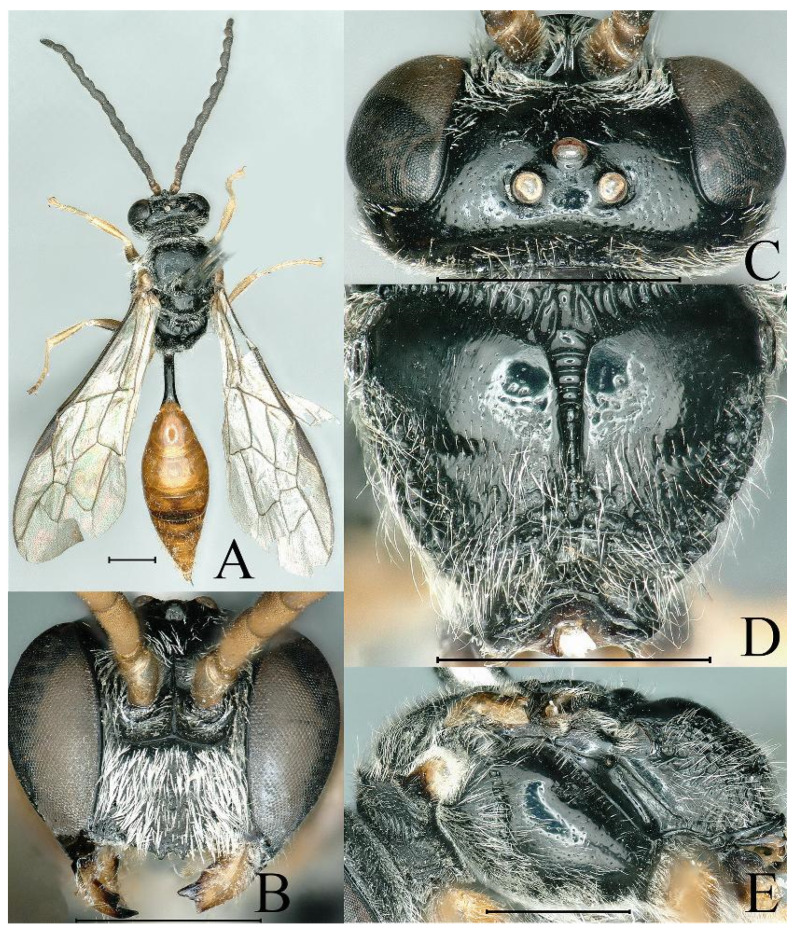
*Psenulus clypeoconvexus* Jiang & Ma, sp. nov., male, paratype. (**A**) Habitus, dorsal view; (**B**) head, frontal view; (**C**) head, dorsal view; (**D**) propodeum, dorsal view; (**E**) thorax, lateral view. Scale bar = 1 mm.

Distribution. China (Zhejiang, Guangxi).

Etymology. The name *clypeconvexus* is derived from the Latin stem *clype*- (=clype) and Latin word *convexus* (=convexus), referring to the central part of the anterior margin of clypeus distinctly prominent.

Notes. The forewing venation is stable in this species. It belongs to the *Psenulus quadridentatus* group.

### 3.3. Newly Recorded Species from China

#### 3.3.1. *Psenulus bicinctus* Turner, 1912 (Figure 5 and Figure 6)

*Psenulus bicinctus* R. Turner, 1912: 363 [17] (Syntypes: ♀, India: Assam: Shillong, BMNH); van Lith, 1962: 110 [13], 1972: 159 [18], 1976: 90 [19].

Material examined. 39♀♀2♂♂, China: Yunnan, Lijiang, Baisha Town, 26°57′ N, 100°12′ E, alt. 2503–2507 m, 25.VIII.2003, coll. Tingjing Li, Qian Jiang, and Zhenshan Geng (YNAU); 2♀♀, China: Yunnan, Baoshan, Longyang District, Pumanshao Village, Lujiang Township, 17.VII. 2006 (1♀), 21.VII. 2006 (1♀), coll. Rui Zhang (YNAU); 1♀, China: Yunnan, KunMing, 15.VII. 1988, coll. Xuexin Chen (ZJU); 1♂, China: Yunnan, Nujiang, Fugong County, Yaping Village, 27.V.2007, coll. Xingyue Liu (YNAU); 1♂, China: Yunnan, Tengchong, Mangbang Town, Taipingpu Village, 23.V.2009, coll. Junhao Huang (YNAU); 1♀, China: Yunnan, Nujiang, Gaoligong Mountains, Yaojiaping Nature Reserve, 25°58′ N, 98°42′ E, alt. 2480 m, 15–30.V.2020, coll. Lang Yi (YNAU); 1♂, China: Gansu, longnan, dangchang County, Daheba Forest Park, alt. 2003 m, 30.VII.2004, coll. Qiong Wu (YNAU); 1♀, China: Shanxi, Hanzhong, Liuba County, Miaotaizi, 28.VII.1982, coll. Jianhua Wei (NWAFU); 1♀, China: Shanxi, Hanzhong, Liping National Forest Park, alt. 1742 m, 22.VII.2004, coll. Min Shi (YNAU); 2♂♂, China: Hubei, Shennongjia, Qianjiaping, alt. 1700 m, 26.VIII.1982, coll. Junhua He (ZJU).

Distribution. China (Gansu, Shanxi, Hubei, Yunnan), India, Nepal, Myanmar.

Notes. Additional description of species: Female (*n* = 46), body length 7.0–8.1 mm; male (*n* = 8), body length 6.3–6.7 mm; mandible bidentate at apex, with distinct inner tooth at base in female, without inner tooth in male (Figure 5B and Figure 6B); frons below interantennal carina with circular depression, distinct or indistinct (Figure 5B); gena near base of mandible with coarse rugae; occipital carina reaching hypostomal carina; mesopleural suture foveolate; ventral surface of female fore and mid trochanters and fore femur basally truncate, ill-defined; inner spur of hind tibia shorter than basitarsus (Figure 5A and Figure 6A); first recurrent vein received by first submarginal cell, second submarginal cell, or interstice between first and second submarginal cells; male antennae with tyloids, segments 3–11 or 3–12 with long, oblique, narrow carina-like tyloids, on segment 12 or 13 reduced to very short carina or point (Figure 6E). This species belongs to the Psenulus rufobalteatus group.

**Figure 5 insects-16-00432-f005:**
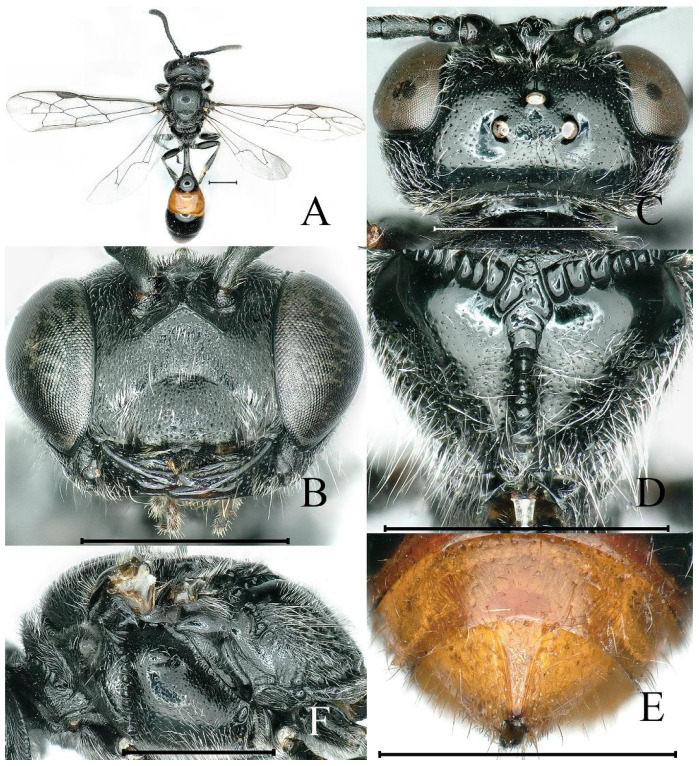
*Psenulus bicinctus* Turner, 1912, female. (**A**) Habitus, dorsal view; (**B**) head, frontal view; (**C**) head, dorsal view; (**D**) propodeum, dorsal view; (**E**) pygidial plate, dorsal view; (**F**) thorax, lateral view. Scale bar = 1 mm.

**Figure 6 insects-16-00432-f006:**
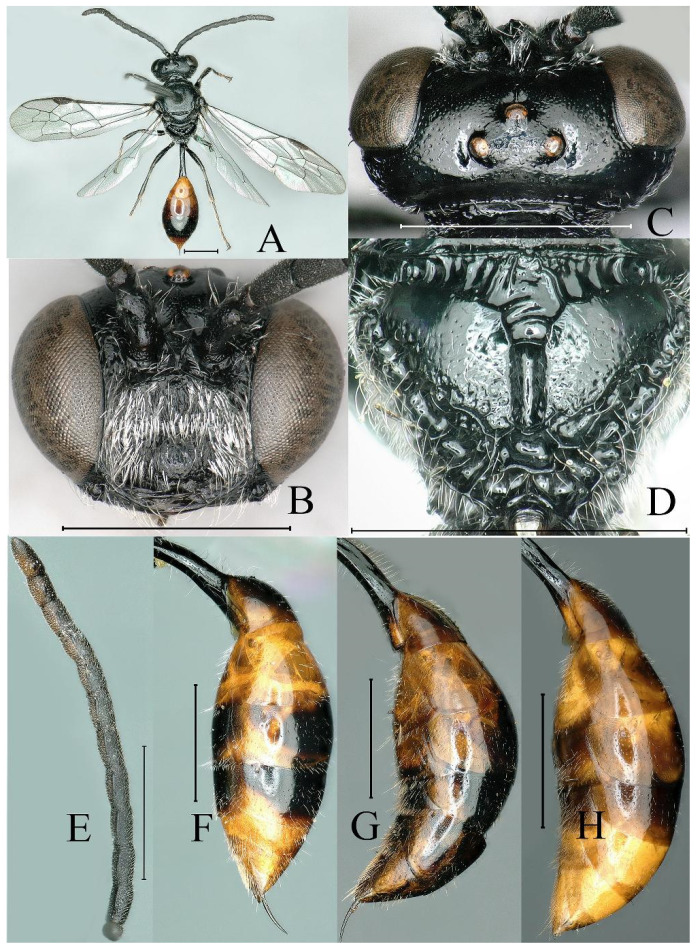
*Psenulus bicinctus* Turner, 1912, male. (**A**) Habitus, dorsal view; (**B**) head, frontal view; (**C**) head, dorsal view; (**D**) propodeum, dorsal view; (**E**) antenna, lateral view; (**F**–**H**) gaster, lateral view. Scale bar = 1 mm.

The following aspects of this species taxonomy need to be clarified and discussed:

This species can be easily recognized by the red colour of gastral segments II, V, and VI, the rest being black [13,17]. However, van Lith observed that the gaster colour was somewhat variable in a female collected from Nepal [18] and in four males from Myanmar [19]. Specimens examined in this study also vary in the gastral colour: in two females from Shanxi Province, gastral segments II, V, and VI are rufous to red, segments I, most of III, and IV are pale brown to dark brown. The males are highly variable in colour, as demonstrated in Figure 6F–H.

*Psenulus bicinctus* Turner, 1912 and *Psenulus rufobalteatus* (Cameron, 1904) [20]. The type localities of *P. bicinctus* (Syntypes: ♀, India: Assam: Shillong, BMNH) and *P. rufobalteatus* (Holotype or syntypes: ♀, India: Assam: Khasia Hills, OXUM) are geographically close, and their morphological characteristics are highly similar based on the descriptions and figures in the literature, the main differences being the body size and gastral segments with red colouration. Van Lith examined the holotypes of both species and considered them to be closely related, being inclined to consider *P. bicinctus* as a subspecies of *P. rufobalteatus* or to synonymize them [18]. Due to the limited material available at that time, it was impossible to determine whether the differences in gastral colour were interspecific or intraspecific. Based on the original descriptions of two species [17,20], the identification and descriptions by van Lith [13,18,19,21], and the materials examined in this study, we think that *P. bicinctus* and *P. rufobalteatus* are one species. Since the holotypes of both species were not available for examination, this revision opinion, however, is not formally proposed.

#### 3.3.2. *Psenulus orinus* van Lith, 1973 (Figure 7 and Figure 8)

*Psenulus orinus* van Lith, 1973:131 [22] (Holotype: ♀, Nepal: 28°00′ N, 85°00′ E, CNC), 1976: 90 [19].

Material examined. 1♀, China: Henan, Jigong Mountain, alt. 700 m, 11.VII.1997, Chikun Yang (CAU); 1♀1♂, China: Guizhou, Fanjing Mount, Huguo Temple, alt. 1300 m, 1.VIII.2001, coll. Meihua Piao and Yun Ma (ZJU); 1♂, China: Guizhou, Fanjing Mount, Huixiangping, 12.VII.1993, coll. Zaifu Xu (ZJU); 1♂, China: Yunnan, Dali, Cangshan Mountain, 4.VI.2009, coll. Jiangli Tan (YNAU); 1♂, China: Yunnan, Tengchong, 1–9.VIII.2011, coll. Jujian Chen (YNAU).

Distribution. China (Henan, Guizhou, Yunnan), Philippines.

Notes. Additional description of species characteristics: Female (*n* = 2), body length 6.5–7.5 mm; male (*n* = 4), body length 6.5–6.8 mm; mandible bidentate at apex, with distinct inner tooth at base in female, without inner tooth in male (Figure 7B and Figure 8B); gena near base of mandible with coarse rugae; mesopleural suture narrowly foveolate along entire length; ventral surface of fore and mid trochanters and fore femur at base truncate, inconspicuously defined in female; inner spur of hind tibia shorter than basitarsus (Figure 8A); first recurrent vein varying in specimens examined: received by the first submarginal cell, or interstice between first and second submarginal cells in females, only received by the first submarginal cell in males (Figure 7A and Figure 8A); antennal segments 3–11 or 3–12 of male with long, oblique, narrow, carina-like tyloids, on segment 12 or 13 reduced to very short carina or point (Figure 8E). This species belongs to the Psenulus rufobalteatus group.

**Figure 7 insects-16-00432-f007:**
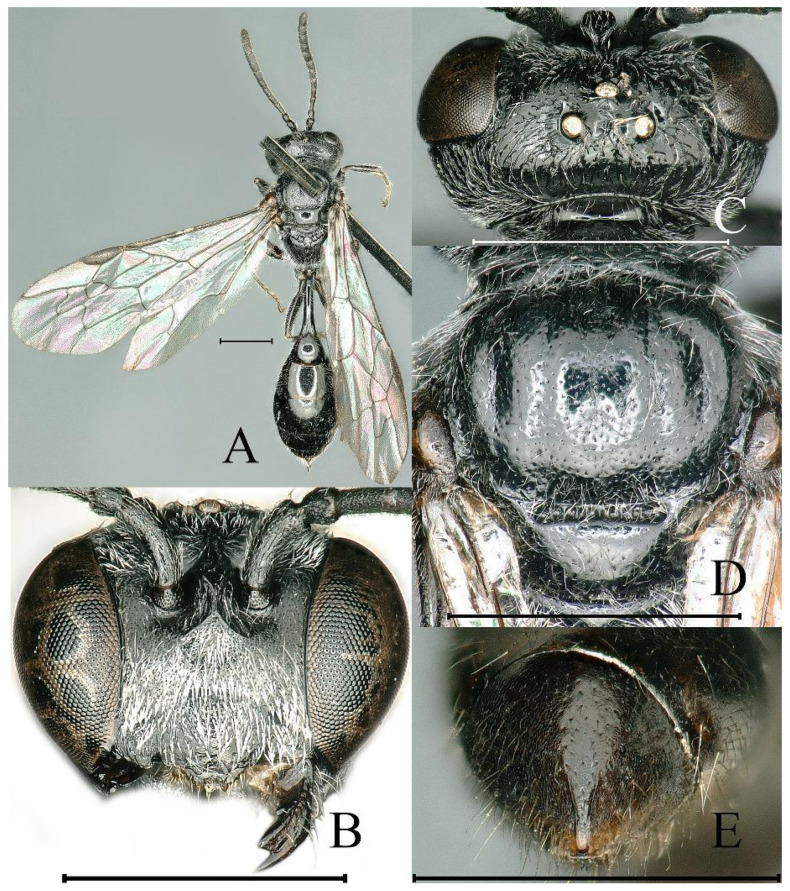
*Psenulus orinus* van Lith, 1973, female. (**A**) Habitus, dorsal view; (**B**) head, frontal view; (**C**) head, dorsal view; (**D**) thorax, dorsal view; (**E**) pygidial plate, dorsal view. Scale bar = 1 mm.

**Figure 8 insects-16-00432-f008:**
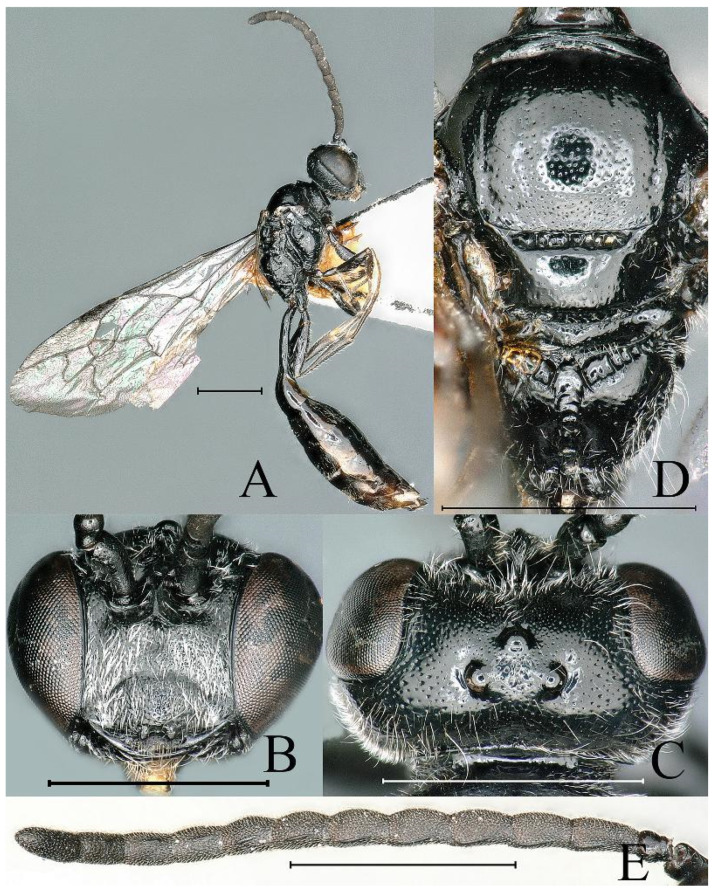
*Psenulus orinus* van Lith, 1973, male. (**A**) Habitus, lateral view; (**B**) head, frontal view; (**C**) head, dorsal view; (**D**) thorax, dorsal view; (**E**) antenna, lateral view. Scale bar = 1 mm.

## 4. Discussion

In the descriptions of *Psenulus*, the identification of red and yellow varies among individuals, and the descriptions of colours as intermediate between red and yellow are even more varied. In this paper, we describe the gastral colour of *P. carinitibialis* as reddish-yellow (Figure 1A) and that of *P. bicinctus* as red (Figure 5A).

There are three situations regarding the reception of the first recurrent vein. It can be received by the first submarginal cell, by the second submarginal cell, or by the interstice between the first and second submarginal cells. We have found that when the first recurrent vein received by the interstice between the first and second submarginal cells, if there is sufficient material, the reception of the first recurrent vein often presents at least two situations. We discussed this characteristic in relation to the species in this study and found it to be either stable or unstable. The applicability of this characteristic in the species identification was discussed.

The genus *Psenulus* plays a crucial role as a predator of parasitic insects, targeting both nymphal and adult stages of various insect groups, including Aphididae, Cicadellidae, Psyllidae, and Delphacidae. During the long evolutionary process, species of *Psenulus* have developed unique hunting behaviours and adaptability to the ecological environment, playing significant ecological functions in the ecosystem. In this study, we provide the first systematic review of the genus *Psenulus* in China, make a significant contribution by describing two new species and providing updated identification keys. Our study lays an essential foundation for global classification.

## Figures and Tables

**Table 1 insects-16-00432-t001:** Structural differences between *P. carinitibialis* and *P. pendleburyi* females.

*P*. *carinitibialis* Jiang & Ma, sp. nov.	*P. pendleburyi* van Lith, 1962
a Transverse carina below antennae distinct and sharp.	Transverse carina below antennae absent.
b Propodeal enclosure complete and broad.	Propodeal enclosure incomplete, with depressions only along lateral margins.
c Median longitudinal groove almost reaching apex of propodeum.	Median longitudinal groove reaching mid-length of propodeum.
d Mid tibia apex lacking depression.	Mid tibia apex with longitudinal depression.
e Hind tibia near base with longitudinal groove and longitudinal lamina carina along groove.	Hind tibia usual, without groove and carina.
f Pygidial plate elongate-triangular, lateral carinae parallel apically, extending for approximately 1/3 of pygidial plate length.	Pygidial plate elongate-triangular, lateral carinae parallel apically, extending for approximately 1/2 of pygidial plate length.

**Table 2 insects-16-00432-t002:** Structural differences between *P. clypeoconvexus* and *P. carinitibialis*.

*P*. *clypeoconvexus* Jiang & Ma, sp. nov.	*P*. *carinitibialis* Jiang & Ma, sp. nov.
a Petiole dark reddish-brown, almost black (♀♂).	Petiole reddish yellow (♀♂).
b Occipital carina reaching hypostomal carina (♀♂).	Occipital carina not reaching hypostomal carina (♀♂).
c Propodeal enclosure narrower and well-defined posteriorly (♀♂).	Propodeal enclosure distinctly broad, not defined posteriorly (♀♂).
d Transverse carina below antennae absent (♀).	Transverse carina below antennae distinctly sharp (♀)
e Interantennal carina in dorsal view markedly broadened, oval, and with shallower depression (♀).	Interantennal carina relatively narrow dorsally and deeply concave (♀).
f Mid tibia without longitudinal carina or row of spines, but with ovally elongate flat depression apically (♀).	Mid tibia with longitudinal carina, one row of spines, and without apical depression (♀).
g Hind tibia basally with shallow groove (♀).	Hind tibia with deep groove basally (♀).
h Pygidial plate with sharp lateral carinae (♀).	Pygidial plate with blunt lateral carinae (♀).
i Depressed margin of clypeus with subsemicircular emargination (♂).	Depressed margin of clypeus with triangular emargination (♂).
j Propodeum behind enclosure shiny, rugose (♂).	Propodeum behind enclosure smooth and shiny, without rugae (♂).

## Data Availability

All data are available in this paper.

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
