# Peer review of "The Genus Psenulus Kohl, 1897 (Hymenoptera: Crabronidae) in China, with Two New Species and Two New Country Records†"

_insects, 2025, doi:10.3390/insects16040432_

Round 1

Reviewer 1 Report

Comments and Suggestions for Authors

Suggested revisions attached in the document.

Author Response

Comments 1: There are some revision suggestions in the “Title”.

Response 1: Thank you for pointing this out. I have accepted the revision suggestion. Therefore, I have revised to ‘The genus Psenulus Kohl, 1897 (Hymenoptera: Crabronidae) in China with two new species and two new country records’.

Comments 2: There are some revision suggestions in the “Simple Summary” and “Abstract”.

Response 2: Thank you for pointing this out. I have accepted all the revision suggestions.

Comments 3: There are some revision suggestions in the “Introduction”.

Response 3: Thank you for pointing this out. I have accepted all the revision suggestions.

Comments 4: It would be worth mentioning that the adult females of Psenulus are able to produce silk, a rare situation among the sphecid wasps. Here are two references to the silk production: de Melo, 1997 (abdominal silk glands in adult females); Serrão and de Oliveira Campos, 2000:480 (abdominal silk glands in adult females).

Response 4: Thank you for pointing this out, I agree with this comment. Therefore, I have done this in page 1, line 35-41.

Comments 5: There are some revision suggestions in the “Materials and Methods”.

Response 5: Thank you for pointing this out. I have accepted all the revision suggestions.

Comments 6: There are some revision suggestions in the “Results: Systematics”.

Response 6: Thank you for pointing this out. I have accepted all the revision suggestions except followings:

  1. I have not added ‘and’ in page 2, line 87 and page 4, line 170, because ‘and’ is generally not used after a semicolon;
  2. I have not deleted ‘medium-sized’ in page 3, line 125, Body length < 9 mm, can be small-sized or medium-sized, so I prefer to use ‘medium-sized’ to limit the scope.

Comments 7: The above couplet is poorly worded. The reader would not be able to tell the difference between “sharply elevated” and “prominent”. Better wording is needed.

Response 7: Thank you for pointing this out. I agree with this comment. Therefore, I have changed ‘Interantennal carina sharply elevated’ to ‘Interantennal carina prominent, dorsally sharp’ in page 2, line 88.

Comments 8: You need either a reference to an illustration here, or a specific proportions of the enclosure. Right now, nobody can tell what is broad and what is narrow.

Response 8: Thank you for pointing this out.

  1. P. parvidentatus van Lith (van Lith, 1972: 163): “Enclosed area of propodeum short, triangular, no median longitudinal carina.” There is no figure and more description about enclosure.
  2. P. gibbus Taylor, Barthélémy, Chi, and Guénard (Taylor et al., 2020): “propodeal enclosure well defined with large fovea (Fig. 15D).” There is no more description about enclosure. According to the figure (I provide this figure for illustration): propodeal enclosure complete, narrow triangle or nearly crescent-shaped.

I have not examined any specimens of P. parvidentatus and P. gibbus. Furthermore, description of characters are simple in both species, Thus, I couldn’t give the specific proportions of the enclosure. To make it easier to understand and distinguish, I have changed to ‘propodeal enclosure incomplete or short/ complete and long’ in in page 3, line 125 and 127.

Comments 9: sclerites: socket?

Response 9: Thank you for pointing this out. I would like to provide an explanation regarding this question. I think sclerites or socket both right in here, sclerites more specific, on the other hand, ‘outer side of antennal sclerites’ follows the description by van Lith (1962) and I prefer to use ‘sclerites’.

Comments 10: do you mean with some segments red?

Response 10: Thank you for pointing this out. I accepted the revision suggestion. Therefore, I have revised to ‘Gaster black with some segments red’ in page 4, line 147.

Comments 11: You need an illustration here!

Response 11: Thank you for pointing this out. To make it easier to understand and distinguish, I have changed ‘pygidial plate narrow and long’ to ‘pygidial plate long and narrowly groove-shaped’ in page 4, line 156.

Comments 12: What are you trying to say?

Response 12: Thank you for pointing this out. I would like to provide a figure for illustration (Tsuneki, 1982e: 25). In order to make it easier to understand, I have changed ‘Mandible tridentate include inner tooth’ to ‘Mandible tridentate’ in page 4, line 163 and 165.

Comments 13: There are some revision suggestions in the “Results: Description of New Species”.

Response 13: Thank you for pointing this out. I have accepted all the reviewer’s revision suggestions except followings:

  1. I have not revised ‘hairs’ to ‘setae’, because the pubescence is soft, I think ‘hairs’ is more appropriate;
  2. I have not revised ‘outer surface of mid tibiae with sharp longitudinal carinae medially and posteriorly which margined behind by one row of five to seven rufous stout spines, and two spines at apex’ to ‘outer surface of mid tibia with sharp longitudinal carinae medially, posteriorly with row of five to seven rufous stout spines, and two spines at apex’, considering the reviewer’s suggestion, I have revised to ‘outer surface of mid tibia with sharp longitudinal carinae medially and posteriorly, margined behind by row of five to seven rufous stout spines, and two spines at apex’ in page 9, line 319-320;
  3. I have not revised ‘near base of hind tibiae with longitudinal groove, margined upper part by many rufous tuberculate spines and inner side by longitudinal lamina carina’ to ‘hind tibia near base with longitudinal groove, with many rufous tuberculate spines dorsally and longitudinal lamina carina laterally’, considering the reviewer’s suggestion, I have revised to ‘hind tibiae near base with longitudinal groove and longitudinal lamina carina along groove, on top of groove with many rufous tuberculate spines’ in page 9, line 321-322.

Comments 14: This is uncomprehensible!

Response 14: Thank you for pointing this out. To make it easier to understand and distinguish, I have changed ‘propodeal enclosure uncomplete which reduced to narrow lateral depression’ to ‘propodeal enclosure incomplete, with depressions only along lateral margin’ in page 8, Table 1b, in P. pendleburyi. In addition, I provide a figure of the propodeal enclosure in P. pendleburyi (van Lith, 1962: 39) for illustration.

Comments 15: ‘lateral and apical of posterior surface and lateral surface of propodeum with irregular carinations and rugae’— comment: ??? and This is not understandable.

Response 15: Thank you for pointing this out. Here is an explanation about all kinds of surfaces of propodeum in this study: propodeum include propodeal enclosure, dorsal surface of propodeum, posterior surface of propodeum, lateral surface of propodeum. In addition, posterior surface of propodeum included upper part, apex, lateral area. These morphological terminologies mostly follow van Lith 1962. To make it easier to understand and distinguish, I have changed ‘lateral and apical of posterior surface and lateral surface of propodeum’ to ‘lateral area and apex of posterior surface and lateral surface of propodeum’ in page 9, line 311-312 and page 10, line 339-340 and page 13, line 385-386.

Comments 16: The above paragraph should be transferred to the Diagnosis before the Species Description—first paragraph of “P. carinitibialis Notes”

Response 16: Thank you for pointing this out. I agree with this comment. Therefore, I have done it in page 8, line 271-275.

Comments 17: I strongly believe that clypeoconvexus would be a better name.

Response 17: Thank you for pointing this out. I have accepted the revision suggestion. Therefore, I have revised ‘clypeconvexus’ to ‘clypeoconvexus’in all of the article.

Comments 18: Contrary to what you say, this description is based on several specimens. Examples: Pronotal lobe ivory to pale yellow. Fore and mid legs pale yellow to yellow. Mid femur with brown patches more or less to various degrees. HL=1.9-2.0 mm; HW=2.5-2.6 mm; POD: OOD=16-18: 24-25. Of course, such a variation cannot be observed in the single specimen (holotype). Please correct the description.

Response 18: Thank you for pointing this out. I have accepted all the reviewer’s revision suggestions. In addition, I would like to explain that ‘pronotal lobe ivory to pale yellow’ and ‘fore and mid legs pale yellow to yellow’ were observed in the single specimen, as demonstrated in Figure 3A and 3F.

Comments 19: What do you mean by ‘lower’? Ventral? Basal?

Response 19: Thank you for pointing this out. To make it easier to understand and distinguish, I have revised to ‘frons below antennae conspicuously raised and without distinct transverse carina’ in page 13, line 408-409.

Comments 20: What do you mean by ‘number’? Row? Is ‘inner’ correct?

Response 20: Thank you for pointing this out. Considering the reviewer’s suggestion, I have revised ‘near base of hind tibiae with longitudinal shallow depression, margined upper part by one number of rufous tuberculate spines, and inner surface by with one longitudinal carina’ to ‘hind tibia near base with longitudinal shallow depression and longitudinal carina along depression, on top of depression with few rufous tuberculate spines’ in page 14, line 433-434.

Comments 21: This Note should be combined with the Diagnosis above.

Response 21: Thank you for pointing this out. I agree with this comment. Therefore, I have done it in page 13, line 387-389.

Comments 22: There are some revision suggestions in the “Results: Newly Recorded Species from China”.

Response 22: Thank you for pointing this out. I have accepted all the revision suggestions.

Comments 23: Do you mean: ‘ventral surface of female fore and mid trochanters and fore femur basally truncate’?

Response 23: Thank you for pointing this out. I agree with this comment. Therefore, I have revised ‘ventral surface of fore and mid trochanters, fore femur at base truncate, ill-defined, in female’ to ‘ventral surface of female fore and mid trochanters and fore femur basally truncate, ill-defined’in page 17, line 490-491.

Comments 24: I see no difference between the first and the third condition!

Response 24: Thank you for pointing this out. What I meant to express was that the reception of the first recurrent vein show differences in female and male specimens examined. To make it easier to understand and distinguish, I have revised ‘first recurrent vein of forewing received by first submarginal cell, or interstitial between first and second submarginal cells in two females of material examined, first recurrent vein of forewing only received by first submarginal cells in four males of of material examined’ to ‘first recurrent vein varying in specimens examined: received by the first submarginal cell, or interstice between first and second submarginal cells in females, only received by the first submarginal cell in males’ in page 20, line 541-544.

Comments 25: There are some revision suggestions in the “Discussion”.

Response 25: Thank you for pointing this out. I have accepted all the revision suggestions.

Comments 26: frequent?

Response 26: Thank you for pointing this out. Frequent is not what I intended to say. To make it easier to understand and distinguish, I have changed ‘and the descriptions of colors intermediate between red and yellow are even more diverse’ to ‘the descriptions of colors as intermediate between red and yellow are even more varied’ in page 20, line 550.

Reviewer 2 Report

Comments and Suggestions for Authors

To provide a detailed review of the attached article, I will evaluate it based on the following aspects: scientific content, English language and grammar, structure, and clarity. Below is a comprehensive peer-review report.

Title and Abstract

- The title is clear and informative, accurately reflecting the content of the study.

- The abstract provides a concise summary of the research objectives, findings (two new species, two new records), and methods.

Suggestions for Improvement: The abstract could benefit from a stronger concluding sentence that emphasizes the significance of the findings.

- Consider rephrasing "we clarify the intraspecific morphological variations observed in the species under study" to "we comprehensively analyze and document intraspecific morphological variations."

Introduction

-The introduction provides sufficient background on the genus Psenulus, its taxonomic importance, and its global distribution.

Grammar corrections: - "which include 173 species" → "which includes 173 species"

- Issue: Some sentences are overly long and complex.

- Break down long sentences for improved readability, e.g.:  Original: "In taxonomic study of Psenulus, two new species and two new records from China were discovered, which new species are described and new records are given supplement description."

- Suggested Revision: "This taxonomic study of Psenulus led to the discovery of two new species and two new records from China. Detailed descriptions of these findings are provided."

- It clearly identifies the research gap (lack of systematic taxonomic studies on Chinese species) and justifies the study.

Suggestions for Improvement: A clearer statement of the study's objectives at the end of the introduction would enhance focus.

Revised: "This taxonomic study of Psenulus led to the discovery of two new species and two new records from China. Detailed descriptions of these findings are provided."

Materials and Methods

- The methodology is well-detailed, including specimen collection, morphological observations, and imaging techniques.

- Abbreviations for institutions are clearly defined.

Results

- The results are well-organized into subsections (e.g., systematics, key to species).

- The identification keys are detailed and scientifically valuable.

Your manuscript provides valuable taxonomic comparisons, but the presentation of differential diagnoses requires restructuring to align with standard entomological reporting conventions. Below are specific recommendations to improve clarity and taxonomic utility:

  1. Comparative Data Presentation

Current Issue: Morphological differences are currently scattered in prose, making direct comparisons challenging.

Revision Suggestion:

- Use tabular format for side-by-side comparisons of key characters.

- Always include parenthetical references to the contrasted species (P. pendleburyi in this case) for immediate clarity.

  1. Standardize Terminology

Current Issue: Inconsistent phrasing (e.g., "normal" vs. "specific variations").

Revision Suggestion: Replace vague terms with precise morphological descriptors. For example:

Original: "Hind tibia normal"

Revised: "Hind tibia lacking longitudinal groove and lamina carina (present in the new species)."

  1. Emphasize Diagnostic Characters

Current Issue: Shared traits and differences are not hierarchically prioritized.

Revision Suggestion: List diagnostic differences first (e.g., carinae structures) before non-diagnostic shared traits (e.g., color pattern).

Add a brief summary statement to highlight taxonomically significant characters (e.g., "The combination of a complete propodeal enclosure and modified mid-tibial apex distinguishes this species from all congeners").

  1. Structural Clarity

Current Issue: Key structures (e.g., pygidium shape) lack explicit descriptions.

Revision Suggestion: Include measurements or ratios (e.g., "pygidial plate 1.5× longer than wide") for quantitative comparisons.

  1. Example of Revised Text

Original: "Apex of mid tibia without depression; area near base of hind tibia with longitudinal groove..."

Revised: "Mid tibia apex lacking depression (vs. with longitudinal depression in P. pendleburyi); hind tibia base with longitudinal groove and lamina carina on inner surface (vs. absent in P. pendleburyi)."

Discussion

- The discussion effectively contextualizes the findings within existing literature.

- It addresses intraspecific morphological variations and their implications for taxonomy.

Suggestions for Improvement:

- Expand on the ecological or evolutionary significance of discovering new species in China.

- Discuss potential applications of this research (e.g., biodiversity conservation, pest control).

- Address limitations of the study (e.g., reliance on morphological data alone) and suggest future research directions.

Scientific Content

- The study makes a significant contribution by documenting two new species and providing updated identification keys.

- It aligns with current taxonomic standards.

Comments on the Quality of English Language

Grammar corrections: - "which include 173 species" → "which includes 173 species"

- Issue: Some sentences are overly long and complex.

- Break down long sentences for improved readability, e.g.:  Original: "In taxonomic study of Psenulus, two new species and two new records from China were discovered, which new species are described and new records are given supplement description."

- Suggested Revision: "This taxonomic study of Psenulus led to the discovery of two new species and two new records from China. Detailed descriptions of these findings are provided."

- It clearly identifies the research gap (lack of systematic taxonomic studies on Chinese species) and justifies the study.

Suggestions for Improvement: A clearer statement of the study's objectives at the end of the introduction would enhance focus.

Revised: "This taxonomic study of Psenulus led to the discovery of two new species and two new records from China. Detailed descriptions of these findings are provided."

Author Response

Comments 1: Title and Abstract: Consider rephrasing ‘We clarify the intraspecific morphological variations observed in the species under study’ to ‘We comprehensively analyze and document intraspecific morphological variations’.

Response 1: Thank you for pointing this out. I have accepted the revision suggestion.

Comments 2: Introduction: Grammar corrections: ‘which include 173 species’ → ‘which includes 173 species’.

Response 2: Thank you for pointing this out. I have revised to ‘Psenulus is the largest genus in Pemphredoninae with 173 species and 50 subspecies’ in page 1, line 23-24.

Comments 3: Introduction: Suggested Revision: “This taxonomic study of Psenulus led to the discovery of two new species and two new records from China. Detailed descriptions of these findings are provided.”

Response 3: Thank you for pointing this out. I have accepted the revision suggestion. Therefore, I have revised to ‘This taxonomic study of Psenulus led to the discovery of two new species and two new records from China. The new species are described and the new records are given supplementary descriptions.’ in page 2, line 44-46.

Comments 4: Introduction: Suggestions for Improvement: A clearer statement of the study's objectives at the end of the introduction would enhance focus.

Response 4: Agree. I have, accordingly, added ‘In the present work, we provide the first systematic review of the genus Psenulus in China, for which no comprehensive taxonomic study has been conducted on the Chinese species of Psenulus.’ in page 2, line 42-44.

Comments 5: Results: Your manuscript provides valuable taxonomic comparisons, but the presentation of differential diagnoses requires restructuring to align with standard entomological reporting conventions. Revision Suggestion: use tabular format for side-by-side comparisons of key characters.

Response 5: Agree. I have, accordingly, added ‘Table 1. Structural differences between P. carinitibialis and P. pendleburyi, females.’ in page 8, and ‘Table 2. Structural differences between P. clypeoconvexus and P. carinitibialis.’ in page 13.

Comments 6: Revision suggestions in the “Results”: Replace vague terms with precise morphological descriptors. For example, revise ‘Hind tibia normal’ to ‘Hind tibia lacking longitudinal groove and lamina carina (present in the new species)’, revise ‘Apex of mid tibia without depression; area near base of hind tibia with longitudinal groove...’ to ‘Mid tibia apex lacking depression (vs. with longitudinal depression in P. pendleburyi); hind tibia base with longitudinal groove and lamina carina on inner surface (vs. absent in P. pendleburyi)’.

Response 6: Thank you for pointing this out. I have accepted all the revision suggestions in page 8, Table 1 and page 13, Table 2.

Comments 7: Revision suggestions in the “Results”: List diagnostic differences first (e.g., carinae structures) before non-diagnostic shared traits (e.g., color pattern).

Response 7: Thank you for pointing this out. I have accepted all the revision suggestions in page 8, line 264-270.

Comments 8: Revision suggestions in the “Results”: Add a brief summary statement to highlight taxonomically significant characters (e.g., The combination of a complete propodeal enclosure and modified mid-tibial apex distinguishes this species from all congeners).

Response 8: Thank you for pointing this out. I agree with this comment. Therefore, I have revised to ‘This species is unique in having the following characters combination, which dis-tinguishes it from all congeners: complete and sharp transverse carina below antennae, outer surface of mid tibia with sharp longitudinal carina medially and posteriorly and without depression at apex, hind tibia near base with longitudinal groove and longitudinal lamina carina along the groove’ in page 8, line 271-275.

Comments 9: Revision suggestions in the “Results”: Include measurements or ratios (e.g., pygidial plate 1.5× longer than wide) for quantitative comparisons.

Response 9: Thank you for pointing this out. I agree with this comment. Therefore, I have revised to ‘lateral carinae parallel apically, extending for approximately 1/3 of pygidial plate length (vs. lateral carinae parallel apically, extending for approximately 1/2 of pygidial plate length in P. pendleburyi)’ in page 8, Table 1 f.

Comments 10: Revision suggestions in the “Discussion”: Expand on the ecological or evolutionary significance of discovering new species in China, discuss potential applications of this research (e.g., biodiversity conservation, pest control), address limitations of the study (e.g., reliance on morphological data alone) and suggest future research directions.

Response 10: Agree. I have, accordingly, added ‘The genus Psenulus plays a crucial role as a predator of parasitic insects, targeting both nymphal and adult stages of various insect groups, including Aphididae, Cicadellidae, Psyllidae, and Delphacidae. During the long evolutionary process, species of Psenulus have developed unique hunting behaviors and adaptability to the ecological environment, playing significant ecological functions in the ecosystem. In this study, we provide the first systematic review of the genus Psenulus in China, make a significant contribution by describing two new species and providing updated identification keys. Our study lay an essential foundation for global classification.’ in page 20, line 560-567.